# Morphological Changes in Blood Cells in a Rat Model of Heatstroke: A Pilot Study

**DOI:** 10.3390/jcm11164821

**Published:** 2022-08-17

**Authors:** Toshiaki Iba, Tomohiro Sawada, Yutaka Kondo, Kenta Kondo, Jerrold H. Levy

**Affiliations:** 1Department of Emergency and Disaster Medicine, Juntendo University Graduate School of Medicine, Tokyo 113-8421, Japan; 2Department of Clinical Laboratory Medicine, Juntendo University Urayasu Hospital, Chiba 279-0021, Japan; 3Department of Emergency and Critical Care Medicine, Juntendo University Urayasu Hospital, Chiba 279-0021, Japan; 4Department of Anesthesiology, Critical Care, and Surgery, Duke University School of Medicine, Durham, NC 27705, USA

**Keywords:** blood smear, heatstroke, hyperthermia, lymphocyte, platelet

## Abstract

Despite the increasing threat of heatstroke with global warming, pathophysiologic injury continues to be defined. In addition, morphological changes of the peripheral blood cells in heatstroke have not been well characterized. We evaluated pathophysiologic changes in bone marrow and blood cells in a rat heatstroke model using a 39.5 °C climate chamber. After three hours of incubation, blood and bone marrow samples were collected for morphology, and the direct effects of heat on leukocytes in vitro were evaluated using time-lapse observation. The blood cell count and peripheral/bone marrow smear were examined either in a lethal model (core body temperature exceeded 42.5 °C) or in a sublethal model (<41.5 °C). Significant decreases in platelet and white blood counts occurred in the lethal model (>35% and >20% decreases, respectively) and changes were less in the sublethal model. Platelet clumping with the appearance of large platelets was observed. The neutrophils often demonstrated hyper-segmented nuclei, and lymphocytes showed reactive or blast-like changes. Further, the direct effect of heat on leukocytes noted apoptotic cell death at 41.5 °C, but subsequent necrosis at 43 °C. In summary, our rodent model showed that heatstroke causes platelet aggregation, leukocyte injury, and aponecrotic cell death. Such changes were milder and reversible in sublethal heatstroke. The appearance of immature cells may result from damage to the bone marrow microenvironment. These findings may provide useful information for potential diagnostic and therapeutic considerations.

## 1. Introduction

Recent climate change has had an important impact on global healthcare. Currently, heatstroke due to global warming is increasing worldwide [1]. For example, the risk for heat-related illness presentation when wet bulb globe temperature (WBGT) reaches 32.2 °C (90.0 °F) is 1.93 times greater than 28.6 °C (83.4 °F), and the risk further increases to 2.53 times higher with 40.6 °C (105 °F) compared with 32.8 °C (91.0 °F) [2]. Currently, such hot days are increasingly more common, even in areas outside the tropical zone.

The pathophysiology of heatstroke is thought to be due to upregulated inflammation and coagulation. Heat stress can induce inflammation and activate coagulation in various ways. Immune cells activate inflammasome, produce inflammatory cytokines, release damage-associated molecular patterns (DAMPs), and upregulate the expression of procoagulant factors. The endothelial damage further accelerates the thromboimflammation. Bouchama et al. [3] explained heat can induce cell injury and apoptotic cell death. Necrotic cells release alarmins, such as interleukin (IL)-1, histones, high mobility group protein box1 (HMGB1), and heat shock proteins, which lead to systemic inflammation [4,5]. In addition, tissue factor and phosphatidylserine exposed on the damaged cells and those on extracellular vesicles released into the systemic circulation activate coagulation cascades causing disseminated intravascular coagulation (DIC) [6]. Together, these factors activate neutrophils, monocytes, and platelets via pattern recognition receptors, including Toll-like and protease-activated receptors [7]. Subsequently, the activated immune cells and proinflammatory cytokines injure vascular endothelium and lead to microcirculatory disturbances [3,5,8].

Although hyperthermia can directly damage the cells, the pathophysiology of heat-related illness has not been assessed extensively in the peripheral circulation for blood cell damage. For example, Huisse et al. [9] reported heat-induced leukocyte activation expressed by upregulated integrin, tissue factor expression, and increased production of reactive oxygen species and IL-8, characteristic changes that provide important pathophysiologic insight, but are not helpful clinically. However, a clinically applicable assessment would be helpful. We hypothesized that blood cells show morphological changes under the heat stress. Since studies examining the effects of heat on blood cells are not well defined, we examined the morphological changes of peripheral blood cells in response to heat stress as a potential clinical evaluation tool.

## 2. Materials and Methods

### 2.1. Subjects and Heatstroke Model

The experiment protocols were approved by the Animal Ethics Committee of the Juntendo University (20071). Male Wistar rats, aged 8–10 weeks and weighing 175–220 g, were purchased from the Laboratory Animal Resource Center and grown at 23.0 ± 1.0 °C with a 12-h light/dark cycle. All the experimental rats were provided with standard rat chow and water ad libitum. For the monitoring of heart rate (HR) and blood pressure (BP), a 3 Fr silicon tube was placed in the left carotid artery and the tube was connected to a transducer. The data were monitored and analyzed by Mac-Lab Hemodynamic Recording System (AD Instruments, Bella Vista, NSW, Australia).

Heatstroke was induced by placing the animals in an artificial climate chamber at 39.5 ± 0.5 °C and relative humidity of 40 ± 5% (heatstroke group: n = 9). Rectal temperature was measured continuously by a thermometer. Animals were anesthetized by intraperitoneal injection of sodium pentobarbital (50 mg/kg body weight) before being subjected to heat and were sacrificed at 180 min after the subjection to the heat (heatstroke model, n = 9). Blood samples were collected from the inferior vena cava with the 3.2% trisodium citrate (the ratio of blood: citrate, 9:1), and the bone marrow samples were aspirated from the femur. In another group, rats were taken out from the chamber when the body temperature reached 41.5 °C (approximately 90 min after being subjected to heat), and animals were kept at 24.0 ± 0.5 °C thereafter (moderate heatstroke group, n = 6). In this group blood samples were obtained at 90 min (or when the body temperature reached 41.5 °C) and 180 min after subjection to the heat. In this group, bone marrow samples were aspirated from the femur at 180 min. For the monitoring of arterial pressure, a catheter was placed in the carotid artery. Animals placed in the unheated chamber (control group: n = 6) were treated identically to those in the heatstroke group, and animals were kept in the artificial climate chamber maintained at 24.0 ± 0.5 °C.

### 2.2. Blood Cell Count, Peripheral Blood Smear Stain, and Bone Marrow Smear Stain

For the blood cell count, fresh citrated blood samples were analyzed by an impedance hematology analyzer (PocH-100iV Diff^®^, Sysmex Corp., Kobe, Japan) [10]. For the morphological examination, peripheral blood films were prepared as thin blood smears, which were fixed in methanol, stained in 10% May-Giemsa stain (GS-10^®^, Sigma-Aldrich, Inc., St. Louis, MO, USA), and examined under the light microscopy at a maximum of 1000× magnification using oil immersion. As for the bone marrow smears, aspirated bone marrow specimens obtained from the femur were suspended in saline solution, then fixed in methanol, stained in 10% May-Giemsa stain, and examined under light microscopy in the same manner with a peripheral blood smear.

### 2.3. Time-Lapse Observation of Leukocytes Subjected to Heat In Vitro

Leukocytes obtained from healthy rats were subjected to heat in vitro. Rats were anesthetized, and blood samples were collected from vena cava and diluted with phosphate-buffered saline (PBS) with 0.1% bovine serum albumin (BSA) and 2 mM EDTA. Then, the samples were centrifuged at 600× *g* for 10 min at 8 °C (KITMAN T-24, TOMY, Tokyo, Japan). The plasma fraction was discarded, and the buffy coat was resuspended to the PBS with 0.1% BSA and 2 mM EDTA. Leukocyte-rich samples in 500 μL of medium (Opti-MEM, Thermofisher Scientific, Waltham, MA, USA) were plated on the slide glasses and incubated for five hours at 37 °C. After incubation, the slide glasses were moved to a heat chamber. The temperature was gradually increased to 41.5 °C at 90 min and 43.0 °C at three hours, and the morphological changes of leukocytes were sequentially observed under the Eclipse Pol microscopic system (Nikon Co., Tokyo, Japan) with DAPI (4′,6-diamidino-2-phenylindole) staining. The abovementioned experiment was repeated three times.

### 2.4. Statistical Analysis

Data are expressed as means ± standard deviation. To compare the mean values between either heatstroke or moderate heatstroke groups and the control group, we used Student’s *t*-test. The level of statistical significance was set at *p* < 0.05. All data were analyzed using SPSS software (SPSS^®^ version 21.0, IBM Corp., Chicago, IL, USA).

## 3. Results

### 3.1. Physiological Responses

After 60 min being subjected to heat, the body temperature reached 41.0 ± 0.2 °C and rose to 41.5 ± 0.2 °C at 90 min. At 180 min, the temperature reached 42.8 ± 0.3 °C. The baseline mean arterial pressure was 90.6 ± 8 mmHg, which increased when the body temperature reached approximately 41.5 °C. Blood pressure continued to be increased (120–130 mmHg) until the body temperature was 42.5 °C. In a typical case, the heart rate and blood pressure decreased immediately after the body temperature reached 43.0 °C. (Figure 1) In the case of moderate heatstroke group, the body temperature decreased to approximately 37.5 °C at 180 min, and rats were recovered. Because the rats died when the body temperature exceeded 43.0 °C, the blood samples were drawn at 180 min. In two out of nine animals, since the blood pressure decreased to 105 mmHg, blood samples were obtained before 180 min.

### 3.2. Blood Cell Counts

Compared with the control group, the platelet counts were significantly lower in the heatstroke group (*p* < 0.01). On the other hand, the platelet counts were not significantly different compared with the control, both at 90 min and 180 min in the moderate heatstroke group. Platelet distribution width (PDW), mean platelet volume (MPV), and platelet large dell ratio (P-LCR) were higher in the heatstroke group compared with the control group (*p* < 0.01, 0.05, 0.01, respectively). Meanwhile, PDW and P-LCR were higher in mild heatstroke group at 90 min (0.05, 0.01, respectively) and 180 min 0.05, 0.01, respectively). White blood count was significantly reduced (*p* < 0.01) in the severe heatstroke group compared with the control group, and the decreased cells were lymphocytes (*p* < 0.01) but not granulocytes (Table 1).

### 3.3. Morphological Changes in the Peripheral Blood Cells

#### 3.3.1. Neutrophils in Heatstroke

The typical neutrophil in control animals demonstrated a donut or a sausage-like nucleus (Figure 2a). In contrast, a multilobed nucleus was observed in the neutrophil of severe heatstroke rats (Figure 3a). This hyper-segmented nucleus is also called “a botryoid nucleus,” a term that signifies a grape-like pattern.

#### 3.3.2. Monocytes in Heatstroke

Monocytes in heatstroke animals were often conjugated with platelets, suggesting that there was an interaction between monocytes and platelets (Figure 3b).

#### 3.3.3. Erythrocytes in Heatstroke

Red blood cells were stained heterogeneously; 10% to 20% of red cells in the samples obtained from rats with severe heatstroke were stained in blue-red (Figure 3a,b). This meta staining represents the reticulocytes (polychromatophilic erythrocyte). Reticulocytes were less frequently seen in the control animals (Figure 2a–d).

#### 3.3.4. Lymphocytes in Heatstroke

Lymphocytes, the most abundant leukocytes in rats, showed oval or notched nuclei and narrow cytosol in the control animals (Figure 2b,c). Meanwhile, reactive lymphocytes were frequently seen in peripheral blood smears from the severe heatstroke rats (Figure 3c), whereas they were less common in moderate heatstroke animals and rarely seen in healthy control. Reactive lymphocytes were large with wide cytoplasm, and the nucleus was irregular, angled, scalloped, or cleaved. Multiple nuclear bodies were often recognized. Other than those, blast-like lymphocytes with a large, light-stained, dispersed chromatin pattern with prominent nucleoli were observed. Apart from those changes, severely damaged lymphocytes showed a ruptured nucleus (Figure 3d).

#### 3.3.5. Platelets in Heatstroke

Platelet aggregation was often observed in the samples from the severe heatstroke group (Figure 4a). In contrast, such a scene was not observed in the samples from the control group (Figure 2a–d). In addition, large (smaller than the red blood cell, Figure 4a, arrowhead) and giant platelets (larger than the red blood cell, Figure 4b) were frequently seen together with the platelet aggregates. In the moderate heatstroke model, platelet aggregation was observed but less common compared with the severe heatstroke group at 90 min (Figure 4c). In the blood smear samples taken at 180 min, the platelet clumping was even less common (Figure 4d).

### 3.4. Morphological Changes of the Precursor Cells in Bone Marrow

Normal progenitor cells were observed in the control group. In contrast, severe bone marrow injury was observed in the severe heatstroke group. The nuclear margins of erythroblasts became fuzzy, and the cellular membranes of those cells were unclear. Prolymphocytes were also damaged, and burst cells were often seen (white arrows, Figure 5). Above mentioned cellular damages were less severe in the moderate heatstroke model (Figure 6), and the injuries were generally mild in the differentiated cells.

### 3.5. Time-Lapse Observation of Leukocytes

Leukocytes stopped moving and started to shrink at 40 °C, and some fragmented into apoptotic bodies, suggesting that apoptosis was initiated at 41.5 °C (Figure 7, left). When the temperature reached 42.0 °C, the cytoplasm of apoptotic cells ballooned, and the apoptosis was suspected to turn into necrosis. At 43.0 °C, most of the half-apoptotic leukocytes progressed to complete necrosis (Figure 7, right).

## 4. Discussion

We noted that platelets and white blood cells were significantly decreased in our study, along with the presence of platelet clumping and large and giant platelets in the severe (lethal) heatstroke animals. In addition, the reticulocyte/erythrocyte ratio was increased, neutrophils often demonstrated a hyper-segmented nucleus, and lymphocytes developed reactive or blast-like changes in the nucleus. In the present study, since body temperature was elevated to 42.5 °C and over the set point of the climate chamber, hypothalamic function was likely disrupted, a derangement that may be involved in heatstroke onset. This dysregulated thermo-control could cause vasoplegia, hypotension, and sudden animal death. Of note, the body temperature decreased, and rats survived if the core body temperature did not exceed 41.5 °C, suggesting the critical temperature for irreversible injury was between 41.5 °C and 42.5 °C.

Concomitantly, the platelet counts significantly decreased, changes consistent with DIC and shock, a well-known complication in severe heatstroke. Prolonged prothrombin time, activated partial thromboplastin time, increased fibrin degradation products, and thrombocytopenia are the typical laboratory findings in heat-induced coagulopathy as well as DIC [11]. In a survey by Bruchim et al. [12], more than 80% of the canines who suffered heatstroke showed thrombocytopenia due to vasculitis and platelet aggregation. In our model of severe heatstroke, we found significant thrombocytopenia with a platelet count decrease of over 35% in the severe heatstroke group compared with controls. By contrast, although the platelet count did not decrease significantly, the platelet distribution width and platelet large cell ratio increased in the moderate heatstroke animals, suggesting that the large immature platelets were increased. Bouchama et al. [13] reported an increased number of circulating leukocytes and lymphocytes in heatstroke patients, and the absolute number of lymphocytes correlated with the degree of hyperthermia. They also reported increases in white blood cell counts in their moderately severe primate heatstroke model but decreased in severe cases [14]. In our study, although leukocytosis was not observed in the moderate heatstroke group, leukocytopenia, especially with 20% decrease of lymphocytes, was observed in the severe heatstroke group.

With respect to peripheral blood smear findings, various patterns of blood cells in hyperthermia patients have been reported. Ranheim [15] noted atypical leukocyte morphology with hyper-segmentation of the nuclei, including those of neutrophils, monocytes, and lymphocytes. Leukocytes have reportedly demonstrated a hyper-segmented spoke-like botryoid pattern of the nucleus in humans [16], and Ward et al. [17] described the botryoid neutrophils and hyper-lobulated lymphocytes in normal blood heated to 42.2 °C for 10 min. The incidence is lower, but eosinophils and basophils are also known to present botryoid nuclei [18]. Consequently, the nucleus changes of the neutrophils and other leukocytes have been previously reported in heat-related illnesses [19]. In general, since nuclear segmentation is considered to progress along with cellular maturation, botryoid nuclei are thought to result from the aging of chromatin. However, in heatstroke, the nuclear segmentation is thought to result from nuclear degeneration induced by hyperthermia.

One of the most interesting observations we found was the damage to the lymphocytes. Ruptured lymphocytes that mimic smudge cells (*Gumprecht* shadows) were often seen in our heatstroke model, suggesting that lymphocytes are more vulnerable to heat, which may relate to the progression of systemic organ damage. Hu et al. [20] reported a decrease in total numbers of splenic regulatory T cells after exposure to heat stress which may exacerbate systemic inflammation. In addition to the destruction of lymphocytes, blast-like or reactive nucleic changes in lymphocytes were frequently observed. The rigid and rectangular shapes of nuclei were thought to be the response to heat and inflammation. Further, if the damage was severe, lymphocytes might progress into oncosis (necrosis) [21]. In addition, ruptured nuclei of lymphocytes were observed, suggesting these cells were damaged severely enough and ruptured on the slide glasses. Mastrorilli [19] reported that severe nuclear derangements such as karyolysis and pyknosis in numerous leukocytes, together with the increased apoptotic bodies in the samples obtained from dogs with heatstroke. Since the described cell deaths were less likely to be seen in the moderate heatstroke group, inflammatory cell death might relate to shock and mortality. In our study, the direct effects of heat on leukocytes were also examined in vitro, and the leukocytes showed apoptotic changes at the temperature of 41.5 °C. The cell death style changed to necrosis if the temperature went higher, and most of the cells died in combination with apoptosis and necrosis, namely aponecrosis at 43.0 °C. This type of sequential change was first observed in our study.

In addition to the cellular damage, our study demonstrated the increase of immature cells in the peripheral blood. For example, the ratio of reticulocytes/erythrocytes elevated, and large and giant platelet counts increased. With respect to reticulocyte increases, Aroch et al. [22] also reported nucleated RBCs were increased in dogs with heatstroke. An increase in nucleated red blood cells in peripheral blood (i.e., rubricytosis) can be unique to dogs and has not been noted in human heatstroke patients [23]. In our rat model, rubricytosis was not found; rather, only increased reticulocytes were found.

Regarding platelets, thrombocytopenia is commonly seen in heatstroke, and decreases in platelets are thought to result from vascular clot formation and consumption [24]. Roberts, et al. [14] reported increases in von Willebrand factor, tissue factor on endothelium, and leukocyte-platelet aggregation in a primate model of heatstroke. In our study, platelet aggregation and monocyte-platelet interaction were frequently observed. Thus, it is reasonable to think the presence of large or giant platelets reflects the increased turnover, and the increase of the platelet large cell ratio supports this finding. The possible explanations for the presence of immature cells include supplementation of consumed or dead blood cells in the systemic circulation. Premature platelet precursors, megakaryocytes, are displaced from the bone marrow into the peripheral blood.

Further, immature cells migrated from the bone marrow due to the damage to the microenvironment in bone marrow caused by heat. Although rubricytosis was previously reported, red blood cell counts were not reduced [19]. Finally, heat-induced systemic inflammation stimulates the release of premature cells into circulation. Leukocytosis and thrombocytosis are well known to be induced by inflammatory cytokines and growth factors in sepsis as well as hyperthermia [25]. Similarly, immature cell increases by similar mechanisms. Taken together, the increase of these immature cells is provoked via the direct heat damage to the blood cells as well as the indirect injury due to the upregulated inflammation in the bone marrow microenvironment.

Bone marrow may be the major target in heatstroke and other acute systemic inflammation. Umemura et al. [26] reported the efficacy of transplantation of bone marrow-derived mononuclear cells in reducing organ dysfunction and survival in a rat heatstroke model. The fundamental mechanism of this therapy is a restoration of bone marrow, attenuation of acute inflammation, and maintenance of endothelial function. Besides bone marrow-derived mononuclear cells, various progenitor cells, including hematopoietic stem cells, and umbilical cord blood cells, secrete antiinflammatory proteins and have protective effects against acute inflammation and heat stress [27]. Liu et al. [28] reported that human umbilical cord blood cell transplantation could also attenuate hypothalamic neuronal damage. The effectiveness of stem cell transplantation supports the idea of bone marrow injury by hyperthermia. Although little information is reported evaluating bone marrow injury in heatstroke [29,30], the present study demonstrated severe bone marrow cell injury, especially in the severe heatstroke group that was predominant in both erythroblasts and prolymphocytes. This observation is consistent with the rubricytosis and increased inflammation that occurs in heatstroke.

## 5. Strength and Limitations

With regard to the experimental design, although this study was an animal experiment, randomization and blinding in ARRIVE Essential 10 were not adapted. Since the main purpose was to examine the usefulness of blood smear examination in heatstroke, we decided that randomization and blinding were not necessary. In addition, the number of animals was limited since the morphological changes were highly reproducible.

In the present study, the morphology of the peripheral blood cells and bone marrow cells was examined in combination with the cell count, blood cell parameters, and an in vitro evaluation of leukocytes. As a result, the lymphocyte damage was reflected by lymphocytopenia, and the large platelet increase was consistent with the increase of the platelet large cell ratio. In addition, the cell death style of leukocytes can change depending on the temperature. To the best of our knowledge, this is the first report that examined the peripheral and bone marrow blood cell changes in lethal and sublethal heatstroke rat models.

Our study has some limitations when comparing changes to heatstroke in humans. First, the composition and morphology of rat blood cells are different from humans. The number of platelets is several times higher than that of humans. The lymphocytes are dominant compared with other leukocytes. The nuclear shape of neutrophils is different, and basophils are rarely seen in the peripheral blood. Further, rodents are more susceptible to heat than humans and rats developed shock and died approximately three hours or later after exposure to the ambient temperature of 39.5 °C, and, as a result, we could observe blood cells changes only for a short period. It is critical to consider these differences when comparing our present model to human heatstroke.

## 6. Summary and Conclusions

The increased platelet clumping, presence of reactive or blast-like lymphocytes, and the presence of hyper-segmented neutrophils are the typical characteristics of the peripheral blood smear in severe heatstroke. These changes were irreversible if the core body temperature exceeded 41.5 °C to 42.5 °C. Direct heat injury and indirect injury via bone marrow damage and heat-induced hyperinflammation are thought to be the major mechanisms of the morphological changes in blood cells. In addition, since the microvascular endothelium is susceptible to heat, platelet consumption and extravasation of neutrophils may further accelerate the progression of organ damage. It is noteworthy to consider the effects of hyperthermia expressed on blood smear findings are the mirror images microvasculature and the microenvironment injury of the bone marrow. Although inflammation and coagulation are critical promoters in the pathogenesis of heatstroke, there is no specific clinically available biomarker at present. Blood smear examination is a simple and readably available technique to provide useful information for understanding the pathophysiology of heatstroke. However, further comparisons with patient clinical findings are needed.

## Figures and Tables

**Figure 1 jcm-11-04821-f001:**
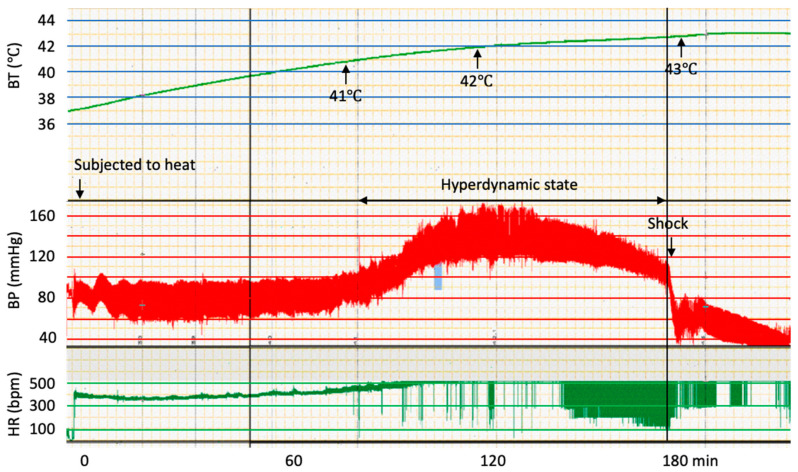
The typical time course of body temperature, blood pressure, and heart rate of a rat subjected to heat. With increasing body temperatures, first a high blood pressure was detected. Then, a hyperdynamic state continued until the body temperature reached 42.5 °C. Eventually, a sudden decrease in blood pressure was observed and animals went into shock when body temperatures approached 43 °C.

**Figure 2 jcm-11-04821-f002:**
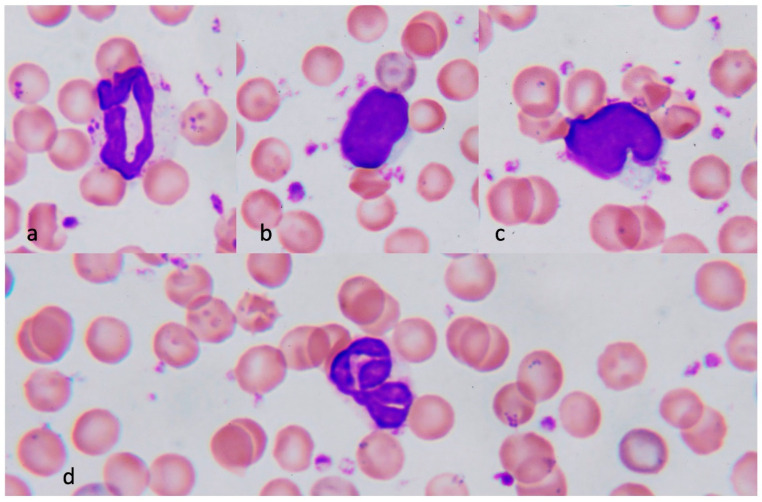
Normal blood cells. Neutrophil shows sausage or donut-shaped nucleus and fine Azur granules in the cytosol (**a**). Lymphocyte has oval (**b**) or notched (**c**) nucleus and small blue-stained cytosol. The nucleus of the neutrophil is sometimes twisted but not hyper-lobulated in normal conditions (**d**). Polychromatophilic erythrocytes are less and platelets are not aggregated.

**Figure 3 jcm-11-04821-f003:**
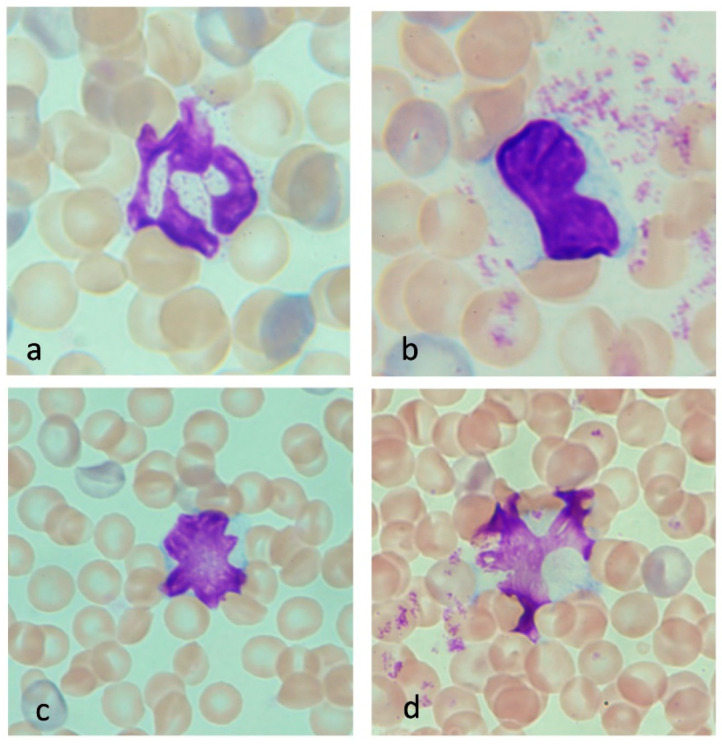
Neutrophils, monocytes, erythrocytes, and lymphocytes in severe heatstroke. A neutrophil from the animal with severe heatstroke showed a multilobed nucleus (**a**). Monocytes with platelet aggregates were frequently seen in the samples from the severe heatstroke rats. The red blood cells were stained heterogeneously (**a**–**d**). Erythrocytes with blue-red color represent the reticulocyte (polychromatic [polychromatophilic] erythrocyte). The lymphocytes from healthy control rats have a round or oval nucleus and narrow cytoplasm. In contrast, lymphocyte from heatstroke rats has scalloped, angle-edged, or cleaved nuclei with nuclear bodies. The lymphocyte has a relatively large cytoplasm and is namely reactive lymphocyte (**c**). Other than that, lymphocytes sometimes showed a blast-like change represented by a large, light-stained, dispersed chromatin pattern with prominent nucleoli. These changes were thought to be induced by the damage of heat. Lymphocytes sometimes showed ruptured nuclei (**d**) or lysed cytoplasmic body that represents oncosis (necrosis).

**Figure 4 jcm-11-04821-f004:**
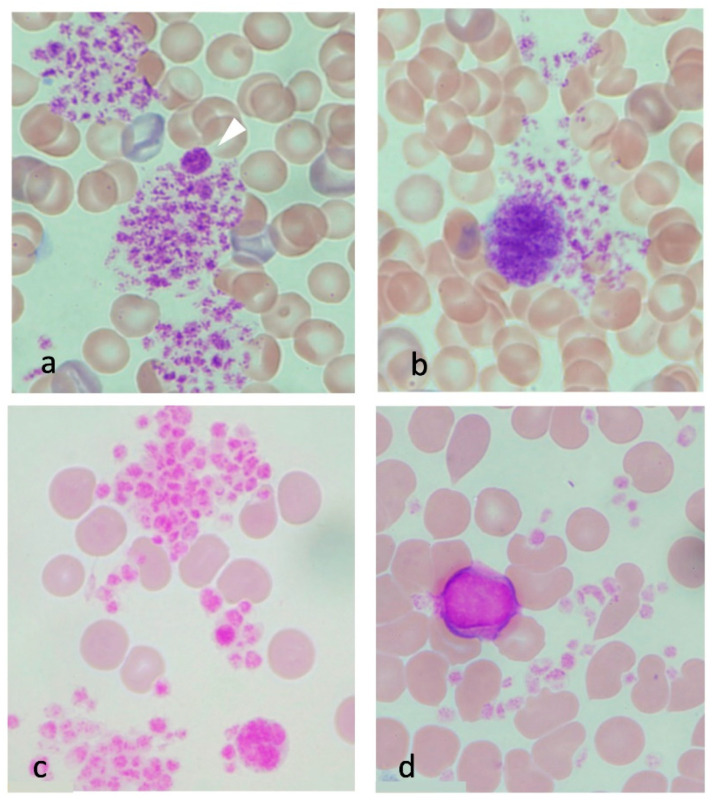
Platelets in severe and moderate heatstroke. Platelet aggregation was not seen in the samples from healthy control rats. By contrast, it was frequently seen in the samples from severe heatstroke rats (**a**). In addition, a large platelet (arrowhead) and a giant platelet are often observed (**b**). The presence of these immature platelets will be the reaction to platelet aggregation. Platelet aggregation was also seen in moderate heatstroke model of rats at 90 min (core body temperature was approximately 41.5 °C, (**c**). The aggregation became less frequent at 120 min when the body temperature returned to 37.5 °C (**d**).

**Figure 5 jcm-11-04821-f005:**
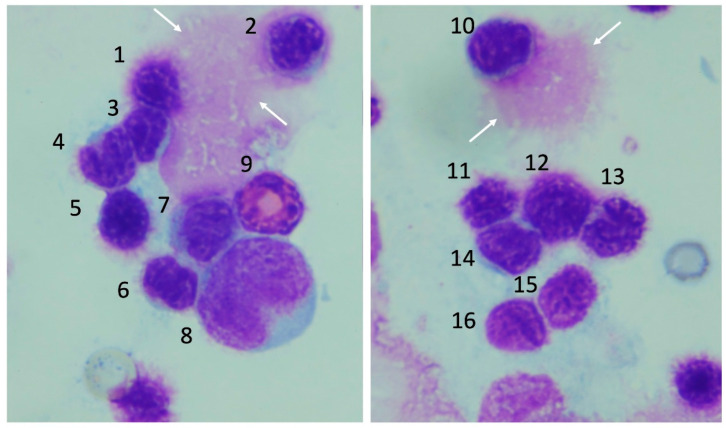
Bone marrow smear in the lethal model of heatstroke. Severe cellular damage was recognized in the bone marrow smear obtained from severe heatstroke group. Burst nuclei are indicated by the white arrows. 1: damaged erythroblast, 2–4 damaged lymphocytes, 5: damaged erythroblast, 6, 7: damaged lymphocytes, 8: promonocyte, 9: eosinophilic band cell, 10: burst lymphocyte, 11–13 damaged erythroblasts, 14–16: lymphocytes.

**Figure 6 jcm-11-04821-f006:**
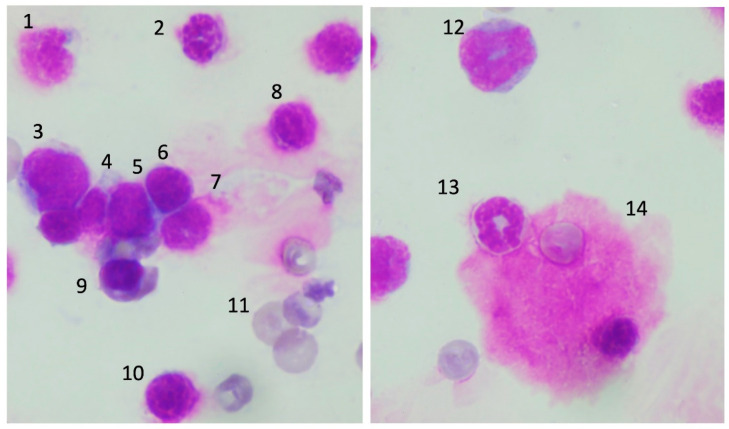
Bone marrow smear in the sublethal model of heatstroke. Bone marrow cells were less damaged in the sublethal model of heatstroke. 1: lymphocyte, 2: neutrophilic band cell, 3: promonocyte, 4–6: lymphocytes, 7: damaged neutrophilic band cell, 8: damaged cell, 9: erythroblast, 10: damaged lymphocyte, 11: reticulocyte, 12: myelocyte, 13: band cell, 14: megakaryocyte.

**Figure 7 jcm-11-04821-f007:**
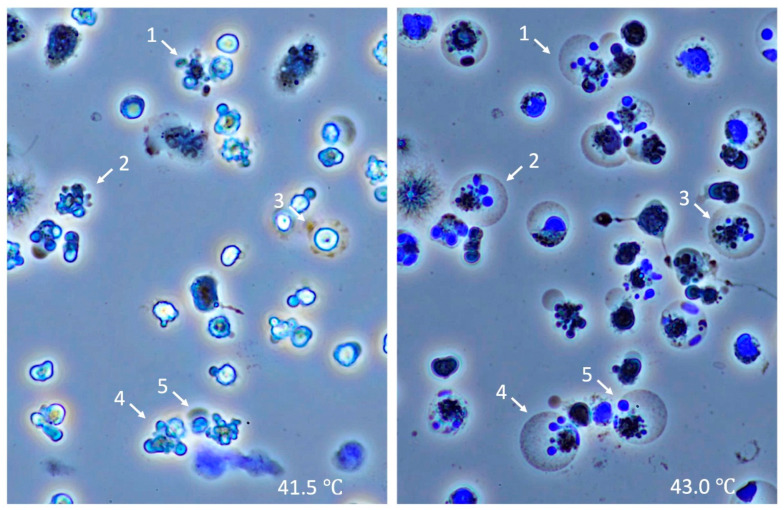
Time-lapse changes of leukocytes subjected to heat. Leukocytes were subjected to heat, and sequential changes were observed under the microscope with DAPI. Leukocytes began to shrink at the temperature of 41.5 °C, and some were fragmented, suggesting apoptosis was initiated at 41.5 °C (**left**). Apoptotic leukocytes started to balloon at 42.0 °C, and at 43.0 °C, most of the leukocytes died in aponecrosis (**right**). The numbers indicate identical leukocytes in different temperatures. DNA was stained blue with DAPI.

**Table 1 jcm-11-04821-t001:** Blood cell count.

	Control Group (n = 6)	Moderate Heatstroke Group, 90 min (n = 6)	Moderate Heatstroke Group, 180 min (n = 6)	Severe Heatstroke Group (n = 9)
WBC count (×10^2^/μL)	68 ± 2	66 ± 1	61 ± 4	58 ± 6 **
Lymphocyte (×10^2^/μL)	53 ± 5	52 ± 5	50 ± 4	42 ± 5 **
Granulocyte (×10^2^/μL)	15 ± 3	14 ± 3	11 ± 3	16 ± 1
RBC count (×10^4^/μL)	714 ± 16	723 ± 18	744 ± 31	785 ± 14 **
Hemoglobin (g/dL)	14.1 ± 2.0	13.5 ± 3.0	14.4 ± 2.9	15.1 ± 2.8
Hematocrit (%)	41.4 ± 6.6	44.2 ± 4.6	45.0 ± 5.1	45.5 ± 7.1
MCV (fL)	62.4 ± 2.4	64.2 ± 2.6	64.2 ± 2.5	62.5 ± 2.3
MCH (pg)	21.7 ± 0.7	21.7 ± 0.6	21.3 ± 0.5	21.1 ± 0.6
MCHC (g/dL)	37.4 ± 1.4	37.2 ± 1.6	37.2 ± 1.0	37.2 ± 1.3
Platelet count (×10^4^/μL)	127.9 ± 18.1	107.5 ± 14.9	116.5 ± 12.1	82.9 ± 21.2 **
PDW (fL)	6.6 ± 0.9	8.4 ± 1.2 *	8.3 ± 0.9 *	8.7± 1.1 **
MPV (%)	6.5 ± 0.8	7.4 ± 1.1	7.4 ± 1.1	7.5 ± 2.1 *
P-LCR (%)	5.5 ± 0.8	8.6 ± 0.9 **	8.3 ± 0.7 **	9.3 ± 1.2 **

WBC: white blood cell, RBC: red blood cell, MCV: Mean Corpuscular Volume, MCH: mean corpuscular hemoglobin, MCHC: Mean Corpuscular Hemoglobin Concentration, PDW: platelet distribution width, MPV: mean platelet volume, P-LCR: platelet large cell ratio. *: *p* < 0.05, **: *p* < 0.01.

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
