# Peer review of "Morphological Changes in Blood Cells in a Rat Model of Heatstroke: A Pilot Study"

_jcm, 2022, doi:10.3390/jcm11164821_

Round 1

Reviewer 1 Report

Iba and colleagues have investigated the morphological changes of blood cells in a rat model of heat-stroke. The authors should be complimented for their efforts in dissecting the effects of heatstroke in a rat model, a highly relevant topic nowadays. Although of interest I do have major points concerning the methods and results and some minor points that could be addressed before this paper can be considered for publication in Journal of Clinical Medicine. 

Major points

Methods

1.     Sample size analysis, randomization, and statistical analysis (i.e. ARRIVE guidelines): 

1a. Was there a sample size analysis conducted? If so, please add this to the paper or explain why not. What was the estimated effect size of heat stroke on blood cells? 

1b. This study would be suited for a randomized design. Why did you not randomize your study and why are there unequal numbers of animals in the severe heatstroke study arm? (in broader sense, animal studies should be planned according to ARRIVE guidelines and preferably preregistered with trial number). Did you plan this experiment using ARRIVE guidelines?

1c. It is unclear to me why the authors justify to use mean and standard deviation in this relatively small exploratory animal study?

1d. It also unclear why a student t-test was used in multiple group comparisons. Why not use a non-parametric test with multiple comparisons? Whether you should correct or not depends on the nature of your study. In this exploratory nature, you could argue not to correct for multiple testing. 

Results

2.     Results; interpretation and functional assays

2a I do not completely understand what was done with animals died before the endpoint was reached. It seems to me that if an animal would die because of heatstroke before the endpoint, you cannot evaluate their whole blood (due to necrosis/apoptosis and a lack of flow initiating coag/inflamm reactions). Just before they collapse, would also make timing between animals incomparable. Please clarify this part better in your text and in future reconsider including animals nearly dying for other outcomes than mortality. To make this point more clear, please add blood continuous blood pressure and mortality in a figure. Also define in this figure when you decided to draw their blood due to near collapse (because this could be seen as mortality).

2b. Why did you choose to only analyse cell count and blood smears/qualitative figures on different blood cells/platelets? This paper begs for a more in-depth analysis using flowcytometry or other tools to further quantify the differences between these groups. I would strongly recommend adding this. 

2c. Functional measures of these cells are lacking (e.g. inflammation, coagulation?). Of course, you observe that during heatstroke cells are destroyed, but in line with my previous point, I would suggest adding some more data on coagulation or inflammation pathways. E.g. are these animals (due to their damaged cells) more hyper- or hypocoaguable? Do you also quantitively measure more platelet-leukocyte aggregates in flowcytometry? 

Minor points

Introduction

-Could you please define a hypothesis 

-Some parts of your introduction need some more explanations (e.g. what is the causal relation between damaged cells (due to heatstroke) and hyperactive (innate) immune cells and alterations in the coagulation system)

Methods

-Line 70-73 What was the diet of the animals prior to the experiment and did they have access to water ad libitum? 

-line 79: 3.2% di- or trisodium citrate? What was the ratio of citrate to blood ïƒ  9:1 parts? 

-line 86: What is the material and size of your arterial cath? How do you record arterial pressure? Using labchart/powerlab system?

-line 97: What type/brand of microscope are you using?

-line 104: To what extend where your blood samples diluted?

-line 106: what type/brand of centrifuge are you using? What are the acceleration and brake settings? This may impact your cells. 

-line 108: What type of medium?

-line 119: Please add the version of SPSS

Results

-as mentioned in major revision points: clarify line 131-132. 

-rephrase line 135 -137 or separate in two sentences.

-line 139 to 141: add your comparison in text ïƒ  higher than ...?

-line 141-142 also here you have to deduce that it is the comparison of severe heatstroke vs control

Discussion 

Line 265-267: here you also hint towards some interesting coagulation assays you can measure in your animals

Overall, blood smears due to their qualitative nature have inter-observer variability and for the clinical translation often a quantifiable measure is easier to use. Why did you choose blood smears? I understand the exploratory nature of understanding a disease, but in my opinion, it is not suited for an easy-to-use tool in clinical practice. Please add a little more on why you think blood smears would be suited to use in clinical practice instead of using more advanced techniques. 

Author Response

Reviewer 1

Major points

Methods

  1. Sample size analysis, randomization, and statistical analysis (i.e. ARRIVE guidelines): 

1a. Was there a sample size analysis conducted? If so, please add this to the paper or explain why not. What was the estimated effect size of heat stroke on blood cells? 

Reply

We thank the reviewer for the comment. Among ARRIVE Essential 10, randomization and blinding were not adapted in this study. Since the main purpose of this study was to explore the usefulness of blood smear examination in heatstroke, and the morphological findings were highly reproducible, we waived randomization and blinding and can restrict the number to nine animals in the severe heatstroke group. We added the above description in the Discussion.

1b. This study would be suited for a randomized design. Why did you not randomize your study, and why are there unequal numbers of animals in the severe heatstroke study arm? (in broader sense, animal studies should be planned according to ARRIVE guidelines and preferably preregistered with trial number). Did you plan this experiment using ARRIVE guidelines?

Reply

This was an exploratory study. In the original protocol, the number of animals was nine in each group. However, the study procedure was very simple, it was an observational study and not interventional, and the results were highly reproducible. Therefore, we decided we could reduce the number of animals in moderate severity and control groups.

1c. It is unclear to me why the authors justify to use mean and standard deviation in this relatively small exploratory animal study?

1d. It also unclear why a student t-test was used in multiple group comparisons. Why not use a non-parametric test with multiple comparisons? Whether you should correct or not depends on the nature of your study. In this exploratory nature, you could argue not to correct for multiple testing. 

Reply

Basically, this study was a qualitative study and not a quantitative one. It has been widely recognized that we cannot diagnose heat-related illness or heatstroke by blood cell count. In this study, blood cell count provided supportive data for the findings in the blood smear. Since the deviation was small and this was an exploratory study, we left them as mean values, and instead, we added “a pilot study” to the title. Regarding the comparison between the groups, the morphological changes in leukocytes were minor, and changes in platelets were reversible in moderate heatstroke, which was reflected in the cell counts. Therefore, the purpose of statistical analysis was mainly the comparison between severe heatstroke and control.

We understand that the methodology is critical and your comments are reasonable, but we would like the reviewer to understand that this report is an exploratory study and there hasn’t been a similar report previously published.

Results

  1. Results; interpretation and functional assays

 2a I do not completely understand what was done with animals died before the endpoint was reached. It seems to me that if an animal would die because of heatstroke before the endpoint, you cannot evaluate their whole blood (due to necrosis/apoptosis and a lack of flow initiating coag/inflamm reactions). Just before they collapse, would also make timing between animals incomparable. Please clarify this part better in your text and in future reconsider including animals nearly dying for other outcomes than mortality. To make this point clearer, please add blood continuous blood pressure and mortality in a figure. Also define in this figure when you decided to draw their blood due to near collapse (because this could be seen as mortality).

Reply

We appreciate your important comment. We added Figure 1, which shows the typical time course of body temperature, blood pressure, and heart rate of a rat subjected to heat. After arterial pressure reached the peak level, the pressure started to decrease gradually. The pressure could decrease suddenly when the mean pressure approached 100 mmHg. We did not use the sample from the dead animals but obtained samples before they went into shock. We added “In two out of 9 animals, since the blood pressure decreased to 105 mmHg, blood samples were obtained before 180 min” in the Result section.

2b. Why did you choose to only analyze cell count and blood smears/qualitative figures on different blood cells/platelets? This paper begs for a more in-depth analysis using flowcytometry or other tools to further quantify the differences between these groups. I would strongly recommend adding this. 

Reply

We appreciate the reviewer’s advice. As the reviewer indicated, we continue the experiment to examine the mechanisms of cell death using immunohistochemical stains and measuring cell death-related biomarkers. However, the primary objective of this study was to examine the usefulness of a blood smear test. This widely used test is available in every hospital, does not require special equipment, and physicians can rapidly evaluate the results. In addition, it will become possible to prepare and analyze the specimens automatically.

2c. Functional measures of these cells are lacking (e.g. inflammation, coagulation?). Of course, you observe that during heatstroke cells are destroyed, but in line with my previous point, I would suggest adding some more data on coagulation or inflammation pathways. E.g. are these animals (due to their damaged cells) more hyper- or hypocoaguable? Do you also quantitively measure more platelet-leukocyte aggregates in flowcytometry? 

Reply

As you indicated, both inflammation and coagulation are the major factors that deteriorate heat illness. Previous studies have reported the usefulness of measuring IL-6 and D-dimer, however, these biomarkers are not heatstroke specific, and it was difficult to determine the cutoffs. Meanwhile, a blood smear test is expected to be useful to discriminate heatstroke from other heatstroke-mimicking diseases such as sepsis and COVID-19. The increase of heatstroke-mimicking sepsis and the coexistence of heatstroke and COVID-19 are the current issues.

Minor points

Introduction

-Could you please define a hypothesis 

Reply

We added “We hypothesized that blood cells show morphological changes under the heat stress” in the text.

-Some parts of your introduction need some more explanations (e.g. what is the causal relation between damaged cells (due to heatstroke) and hyperactive (innate) immune cells and alterations in the coagulation system)

Reply

We added “Heat stress can induce inflammation and activate coagulation in various ways. Immune cells activate inflammasome, produce inflammatory cytokines, release damage-associated molecular patterns (DAMPs), and upregulate the expression of procoagulant factors. The endothelial damage further accelerates the thromboimflammation” in the text.

Methods

-Line 70-73 What was the diet of the animals prior to the experiment and did they have access to water ad libitum? 

Reply

We added “All the experimental rats were provided with standard rat chow and water ad libitum” in the text.

-line 79: 3.2% di- or trisodium citrate? What was the ratio of citrate to blood à 9:1 parts? 

Reply

We added “trisodium citrate (the ratio of blood: citrate, 9: 1)” in the text.

-line 86: What is the material and size of your arterial cath? How do you record arterial pressure? Using labchart/powerlab system?

Reply

We added “For the monitoring of heart rate (HR) and blood pressure (BP), a 3 Fr. silicon tube was placed in the left carotid arteryand the tube was connected to a transducer. The data were monitored and analyzed by Mac-Lab Hemodynamic Recording System (AD Instruments, NSW, Australia)” in the text.

-line 97: What type/brand of microscope are you using?

Reply

We added “Eclipse Pol microscopic system (Nikon Co., Tokyo, Japan)” in the text.

-line 104: To what extend where your blood samples diluted?

Reply

We diluted 1 mL of blood samples to 3 mL.

-line 106: what type/brand of centrifuge are you using? What are the acceleration and brake settings? This may impact your cells. 

Reply

We used KITMAN T-24 (TOMY, Tokyo, Japan).  It is a standard method for plasma separation.

-line 108: What type of medium?

Reply

We used Opti-MEM (ThermoFisher Scientific, Waltham, MA, USA).

-line 119: Please add the version of SPSS

Reply

We used version 21.0.

Results

-as mentioned in major revision points: clarify line 131-132. 

Reply

We added, “In two out of 9 animals, since the blood pressure decreased to 105 mmHg, blood samples were obtained before 180 min.”

-rephrase line 135 -137 or separate in two sentences.

Reply

The sentence was divided into two sentences.

-line 139 to 141: add your comparison in text à higher than ...?

Reply

We added “compared to the control group” in the text.

-line 141-142 also here you have to deduce that it is the comparison of severe heatstroke vs control

Reply

We added “in the severe heatstroke group compared to the control group” in the text.

Discussion 

Line 265-267: here you also hint towards some interesting coagulation assays you can measure in your animals

Reply

Although inflammation and coagulation are critical promoters in the pathogenesis of heatstroke, there is no specific clinically available biomarker at present. We added the above description to the Summary and Conclusion.

Overall, blood smears due to their qualitative nature have inter-observer variability and for the clinical translation often a quantifiable measure is easier to use. Why did you choose blood smears? I understand the exploratory nature of understanding a disease, but in my opinion, it is not suited for an easy-to-use tool in clinical practice. Please add a little more on why you think blood smears would be suited to use in clinical practice instead of using more advanced techniques.

Reply

Veterinarians gave us a hint regarding the diagnosis of heatstroke. The blood smear test is popularly performed for the diagnosis of heatstroke in dogs. The test can be subjective, and that is why we presented the typical findings in this manuscript. We agree that quantitative tests are easier to judge; however, there is no useful biomarker at present. As noted, blood films can be readily evaluated in every local hospital, the test is not an expensive device, and the result can be rapidly determined.

Reviewer 2 Report

This is an important study because recent climate change has had an important impact on global healthcare. Currently, heatstroke due to global warming is increasing worldwide.

The study included a group of 9 and 6 animals which were exposed to heat in the climate chamber, and the control group consisted of 6 rats. The downside is a small study (15 rats) and control group (6 rats). Then, after the incubation period, blood and bone marrow samples were taken for morphology, and direct the effect of heat on in vitro leukocytes was assessed using a time-lapse observation. The authors observed clumping of platelets and morphotic changes in blood cells, such as plateles ans neutrophils. This study provides information on the pathophysiological changes in blood cells following heat stroke.  It raises a very interesting and important topic.

The presented manuscript has a typical structure and is clearly divided. Consists of 1 table and 6 figures with clear and correct descriptions. The discussion is exhaustive, although little research has been done on the subject. The results and conclusions are correct. In the presented manuscript, I did not find any obvious linguistic errors.

Author Response

Reply

The authors appreciate the reviewer’s positive comments.

Round 2

Reviewer 1 Report

Dear Iba and colleagues, 

I appreciate your answers, addition of vital parameters (fig 1) and the corrections you have made. Also, I value that you have highlighted the exploratory and pilot nature of the study, which is important for your readers. I have suggested to accept this manuscript with minor revisions (some grammar issues and I recommend to move the first paragraph of your discussion to limitations).

I would like to congratulate you with your important findings and hope to see more of your work on coagulation and inflammation pathways after heatstroke in the future. 

Minor points:

1.     Figure 1: grammar

The bood pressure rose along with the body temperature elevated. The hyperdynamic state continued until the body temperature reaches 42.5 C°. The blood pressure decrase thereafter and the subject fell into shock when the body temperature approached 43 C°

Please change the subtext in something like this: 

‘With increasing body temperatures, first a high blood pressure was detected. Then, a hyperdynamic state continued until the body temperature reached 42.5 °C. Eventually, a sudden decrease in blood pressure was observed and animals went into shock when body temperatures approached 43 °C’

2.     Also, I would reference figure 1 somewhere in your text. 

3.     In your discussion you start with your main limitation, I would suggest moving that part to your strengths/limitations section. In that way it remains important, but it doesn’t interfere with your main message. 

---

For future studies I would recommend, even for relatively small exploratory studies, to register your trial and to adhere to the ARRIVE guidelines. Also, the sample size calculations are important to describe and the justification of any deviations.

Although you should be complimented that you reduced the number of animals during your study, this point is only known by your research team and it remains unclear for your readers. 

I enjoyed our discussion and I am looking forward to see the published version of this paper. 

Author Response

We thank the Editor and reviewer for the opportunity to revise the manuscript again. We also appreciate reviewer 1 for the careful reading and kind suggestions.

  1. Figure 1: grammar

‘The bood pressure rose along with the body temperature elevated. The hyperdynamic state continued until the body temperature reaches 42.5 C°. The blood pressure decrase thereafter and the subject fell into shock when the body temperature approached 43 C°’

Please change the subtext in something like this: 

‘With increasing body temperatures, first a high blood pressure was detected. Then, a hyperdynamic state continued until the body temperature reached 42.5 °C. Eventually, a sudden decrease in blood pressure was observed and animals went into shock when body temperatures approached 43 °C’

Reply

We appreciate the reviewer for the revision of the figure legend. We changed the legend as the reviewer suggested.

  1. Also, I would reference figure 1 somewhere in your text. 

Reply

 We indicated it in the text (line 138).

  1. In your discussion you start with your main limitation, I would suggest moving that part to your strengths/limitations section. In that way it remains important, but it doesn’t interfere with your main message. 

Reply

We appreciate your advice. The sentences were moved to the Strength and Limitation section accordingly.

Finally, the authors thank you again for your friendly advice for future studies.